# Herbivory Amplifies Adverse Effects of Drought on Seedling Recruitment in a Keystone Species of Western North American Rangelands

**DOI:** 10.3390/plants11192628

**Published:** 2022-10-06

**Authors:** Mathew Geisler, Sven Buerki, Marcelo D. Serpe

**Affiliations:** Department of Biological Sciences, Boise State University, Boise, ID 83725, USA

**Keywords:** *Artemisia tridentata* ssp. *wyomingensis*, herbivory, seedling survival, plant water potential, reproductive development

## Abstract

Biotic interactions can affect a plant’s ability to withstand drought. Such an effect may impact the restoration of the imperiled western North American sagebrush steppe, where seedlings are exposed to summer drought. This study investigated the impact of herbivory on seedlings’ drought tolerance for a keystone species in this steppe, the shrub *Artemisia tridentata*. Herbivory effects were investigated in two field experiments where seedlings were without tree protectors or within plastic or metal-mesh tree protectors. Treatment effects were statistically evaluated on herbivory, survival, leaf water potential, and inflorescence development. Herbivory occurrence was 80% higher in seedlings without protectors. This damage occurred in early spring and was likely caused by ground squirrels. Most plants recovered, but herbivory was associated with higher mortality during the summer when seedlings experienced water potentials between −2.5 and −7 MPa. However, there were no differences in water potential between treatments, suggesting that the browsed plants were less tolerant of the low water potentials experienced. Twenty months after outplanting, the survival of plants without protectors was 40 to 60% lower than those with protectors. The percentage of live plants developing inflorescences was approximately threefold higher in plants with protectors. Overall, spring herbivory amplified susceptibility to drought and delayed reproductive development.

## 1. Introduction

The capacity of plants to withstand drought varies at different stages of development [1,2]. Seedlings and juveniles are typically the most vulnerable to drought [1]. At these stages, the lack of an extensive root system markedly limits the plant’s ability to maintain water uptake as the soil dries out [2,3]. As a result, drought stress is a major factor limiting plant recruitment from natural seed banks or seeds and seedlings planted in restoration and reforestation projects [1,4]. The adverse effects of drought on seedling establishment will likely worsen due to the expected increase in the frequency and intensity of drought with climate change [5].

In addition to drought, the seedling stage tends to be more susceptible to other stresses and disturbances than later stages of development [1]. One such disturbance is herbivory [6]. While exceptions exist, there is often an increase in plant chemical and structural defenses from the seedling to the mature stage, making the former more prone to attack by herbivores [6,7]. Furthermore, the limited storage reserves present in seedlings can limit their recovery following herbivory, resulting in low herbivory tolerance [8].

Herbivory and drought may overlap or succeed each other in either order. Because the plants’ requirements to cope with drought and herbivory differ, the effect of one stressor can reduce the plant’s ability to withstand the other [9]. For example, maintaining water uptake during drought is often mediated by the preferential allocation of photosynthates to root growth rather than to plant defenses or shoot growth, thus leading to more susceptibility and less tolerance to herbivory [10]. On the other hand, the loss of leaf area caused by herbivory and the energy and carbon needed to recover from this damage can cause a decrease in non-structural carbohydrates (NSCs) [9,11]. This decrease leaves plants more vulnerable to water stress because NSCs contribute to the growth of fine roots, osmotic adjustment, and sustained metabolism when the stomata close [12,13,14]. However, the impacts of drought and herbivory on plant performance are complex, which can lead to situations when the effect of one stressor alleviates the other. For instance, defoliation caused by herbivory reduces the area for transpiration and, in some plants, increases the reallocation of NSCs to the roots [15]. These effects may partly offset herbivory’s adverse effects on drought tolerance, leading to non-additive responses where the impact of the two stressors is less than the sum of their individual effects [9,16].

Although plants in semiarid and arid ecosystems have different strategies to cope with drought, their natural recruitment or seedling establishment following seeding and outplanting can be very low [17,18]. A semiarid habitat where this has been commonly observed is the sagebrush steppe in western North America [18,19,20]. This habitat covers approximately 450,000 km^2^ and is characterized by a vegetative community composed of perennial grasses, forbs, biological soil crusts, and several subspecies of the shrub *Artemisia tridentata* Nutt (big sagebrush) [21,22]. Over the past century, sagebrush habitats have been disturbed by overgrazing and invasion by exotic grasses [21,23]. In particular, invasion by *Bromus tectorum* has disproportionately affected sagebrush communities by decreasing the fire return interval, which tends to eliminate *A. tridentata* and other components of the native vegetation [22,23]. 

*Artemisia tridentata* plays various ecological roles. It contributes to developing a heterogeneous landscape that provides microsites for understory plant species and habitats and food for local animals [24,25,26]. Due to the critical functions of *A. tridentata* in sagebrush habitats, there is considerable interest in reestablishing this species following fires [27,28]. Attempts to reestablish big sagebrush have been extensive and costly, with varied but generally low success rates [19,29,30]. The cause of this variability is not well understood, but summer drought significantly limits seedling establishment in this species [20,31,32]. 

Because *A. tridentata* is often the only green vegetation for several months, it is an important food source for various animals. These include birds such as the greater sage-grouse (*Centrocercus urophasianus*), large ruminant ungulates such as the pronghorn (*Antilocapra americana*) and mule deer (*Odocoileus hemionus*), and small mammals such as pygmy rabbits (*Brachylagus idahoensis*) [24,33,34]. Although *A. tridentata* accumulates many secondary metabolites to deter herbivores, the damage caused by herbivory can be extensive [35,36]. However, adult *A. tridentata* plants can usually tolerate this damage [35,37,38]. 

Based on results in other species, herbivory tolerance is likely lower for seedlings and young plants of *A. tridentata* than their older counterparts [6,7]. Damage to young *A. tridentata* plants by herbivores is frequently observed, but the impact of this damage on plant survival is not well characterized [18,32]. Ungulates can cause extensive shoot damage to young plants, which tends to be fatal because the plants do not resprout from the roots [28,32]. However, in many sagebrush communities, ungulates are rare [39]. In these communities, herbivory by small mammals or insects may only cause partial damage that, while not directly lethal, may affect the plants’ ability to cope with other stressors and thus indirectly reduce their survival. [9,40]. 

In preliminary experiments, we observed herbivory in 1–2-year-old *A. tridentata* plants by small mammals (presumably ground squirrels) and, to a lesser extent, by harvester ants and grasshoppers. This damage was often not immediately fatal since seedlings resprouted after herbivory. However, as discussed earlier, results in other species have suggested that herbivory might make *A. tridentata* seedlings more susceptible to abiotic stresses, particularly drought [16]. To analyze the plausible effect of herbivory on the seedlings, we placed 10-month-old seedlings of *A.tridentata* ssp. *wyomingensis* (Wyoming big sagebrush, hereafter referred to as *A. tridentata*) under one of three treatments: without tree protectors, within plastic tree protectors, or within metal-mesh tree protectors. Preliminary observations suggested that these treatments would lead to different levels of herbivory. Similar experiments were started in two consecutive years. In both experiments, we evaluated the effect of the treatments on the time course of herbivory and mortality. In addition, we measured variables indicative of plant water status for the second experiment. We hypothesized that in browsed plants, the carbon demand for shoot regrowth would occur at the expense of root growth, reducing the plant’s ability to extract water from deeper soil, resulting in lower plant water potential and higher summer mortality than in uneaten plants. 

## 2. Results

### 2.1. Climatic Conditions during the Experimental Period

Temperature and precipitation followed patterns typical of the area, with most precipitation occurring during the winter and spring (Figure 1). However, there were some differences between the years. In 2020, significant rainfall occurred in late spring and early summer, and soil moisture did not decline as low as in the summer of 2019 (Figure 1B). In addition, the summer of 2021 presented climatic conditions more conducive to drought than in the previous two years. Precipitation in the winter and spring of 2021 was about 40% lower than in the winter and spring of 2019 and 2020. Furthermore, summer temperatures in 2021 were higher than in the previous two years (Figure 1A).

### 2.2. First Field Experiment

The first experiment started in October 2018 in Kuna Butte, ID, USA (43°26′47.32″ N, 116°26′48.61″ W). We outplanted 750 seedlings in a lattice at a distance of about 1.5 m from each other. The seedlings were randomly assigned to one of three treatments (*n* = 250): without tree protector, with plastic tree protector (25.2 mm mesh, 44 cm height, and 10 cm in diameter), or with metal tree protector (6 mm mesh and closed at the top) (Figure 2).

Independent of the protector treatment, damage to or losses of seedlings were minimal during the fall of 2018 (Figure 3). In contrast, significant damage due to herbivory occurred by the late winter of 2019. At this time, the percentage of seedlings that experienced herbivory was about 90%, 7%, and 1% for the no-protector, plastic, and metal protector treatments, respectively (Figure 3A, *p* < 0.0001 between no-protector and the other two treatments). The damage varied between seedlings that experienced herbivory, but a representative example of the observed damage is shown in Figure 3C,D. Subsequently, herbivory damage markedly declined (Figure 3A), and by 17 April 2019, 68% of the injured seedlings had begun to resprout. During the summer, additional herbivory occurred in the vicinity of harvester ant nests. These plants were defoliated entirely and did not recover from herbivory. However, only a few plants were affected, and the loss was similar between treatments. In contrast to the first winter and spring in the field, herbivory during the winter and spring of 2020 was negligible (Figure 3A).

In plants without protectors, mortality occurred during spring 2019 and continued during the summer (Figure 3B). In contrast, mortality primarily happened during the summer in plants with plastic and metal protectors. By the end of summer 2019, survival was 23.4, 75.6, and 85.2% for the no-, plastic, and metal protector treatments, respectively (Figure 3B). Moreover, even though herbivory was minimal during the summer, plants without protectors showed lower summer survival than the other treatments. Starting with the plants that were alive on 29 June 2019, summer survival was 59.1% for plants without protectors, 77.9% for plants within plastic protectors, and 85.5% for plants within metal protectors (*p* < 0.0001 between plants without and with protectors). Subsequently, survival slightly declined for all treatments. At the end of July 2020, survival from the beginning of the experiment was 19.5, 70.7, and 77.5% for the no-, plastic, and metal protector treatments, respectively (Figure 3B). These differences were significant between the no-protector and the other treatments (*p* < 0.0001) but not between the plastic and metal protector treatments (*p* = 0.08).

For the plants alive at the end of July 2020, there were differences in the percentage of plants bearing inflorescences. This percentage was 2.2% in plants without protectors, 23.5% in plants within plastic protectors, and 41.9% in plants within metal protectors. Significant differences were detected between each pair of treatments: *p* = 0.001 for the metal vs. plastic protector comparison, *p* = 5.3 × 10^−7^ for metal vs. no protector, and *p* = 0.0002 for plastic vs. no protector.

### 2.3. Second Field Experiment

This experiment started in October 2019 in a plot adjacent to that used in the previous experiment. We followed identical outplanting methods and protector treatments but with only 150 seedlings per treatment (*n* = 150). The time course of herbivory damage observed in this experiment was similar to that observed following the 2018 outplanting. Damage to or loss of seedlings was minimal during the fall of 2019 and most of the winter of 2020 (Figure 4A). In contrast, significant damage due to herbivory occurred in March 2020. In that month, the percentage of seedlings that experienced herbivory was 85%, 25%, and 5% for no-protector, plastic, and metal protector treatments, respectively (*p* < 0.0001). Herbivory damage markedly declined by April 2020 (Figure 4A), and the injured plants began to resprout. However, these plants did not fully recover in terms of their size. At the end of summer 2020, the projected shoot area in the no-protector treatment was lower than in the other treatments (*p* < 0.0001), with values of 16.03 (±2.05), 45.53 (±4.97), and 49.06 (±4.27) cm^2^ for the no-protector, plastic, and metal protector treatment, respectively. In the summer, a few plants suffered terminal damage from harvester ants, but, other than this damage, herbivory was minimal during the rest of the experiment.

On 1 May 2020, after most herbivory had occurred, survival was similar between treatments (Figure 4B). From then on, however, survival rates began to differ. In particular, seedlings without protectors had lower survival at the end of summer 2020 than those with protectors (*p* < 0.001). This trend continued until August 2021. At this time, the survival of seedlings without protectors was 51.9%, those with plastic protectors was 76.7%, and those with metal protectors was 89% (Figure 4B). These differences in survival were significant between each pair of treatments (*p* = 0.012 for the metal vs. plastic protector comparison, *p* = 2.9 × 10^−9^ for metal vs. no-protector, and *p* = 3.56 × 10^−5^ for plastic vs. no-protector). Additionally, herbivory in early spring reduced the proportion of live plants with inflorescences. In July 2020, this proportion was about three times higher in seedlings with protectors than in those without them (Table 1, *p* < 0.0001). Similar results were observed in July 2021. In this experiment, an additional measure of survival and inflorescence development was made one year later, in July 2022 (Appendix A). Survival remained similar to August 2021, being 49.3% for seedlings without protectors, 74.3% for those within plastic protectors, and 86.9% for seedlings within metal protectors. These differences were significant between each pair of treatments. In contrast, the percentage of live plants bearing inflorescences increased in all treatments, and no statistical differences were noted between them (Table 1).

To characterize the degree of water stress the plants experienced, we measured predawn and midday leaf water potential (Ψ_l_) during the summer of 2020 and the midday Ψ_l_ and stomatal conductance (g_s_) during the spring and summer of 2021. These measurements were only conducted in the no-protector and metal protector treatment because these treatments showed the highest difference in terms of the extent of herbivory. In 2020, predawn Ψ_l_ ranged from −1.4 to −7 MPa, and midday Ψ_l_ ranged from −1.6 to −8 MPa (Figure 5A,B). The variation in Ψ_l_ increased during the progression of the summer. However, except for a day in mid-August, the median values of midday Ψ_l_ remained relatively constant from July to December (Figure 5A). In addition, differences in predawn or midday Ψ_l_ between seedlings without and with metal protectors were not significant.

In 2021, midday Ψ_l_ declined from about −1 MPa in spring to about −2.5 MPa in mid-summer (Figure 5C). As in 2020, differences in Ψ_l_ between the no- and metal protector treatments were not significant. However, there were some differences between the years. Although 2021 was drier than 2020 (Figure 1), Ψ_l_ values were higher in the 2021 summer than in 2020 (Figure 5A,C). Furthermore, for comparable periods, the variability in Ψ_l_ between plants was much less in 2021 than in 2020. Stomatal conductance showed a similar pattern to Ψ_l_, with a decline from spring to summer and no apparent differences between the two treatments (Figure 5D).

## 3. Discussion

This study identified a period about five months after outplanting in which *A. tridentata* seedlings suffered intense herbivory (Figure 3A and Figure 4A). Most seedlings resprouted following this damage. However, herbivory increased the plants’ susceptibility to abiotic stresses, including drought, resulting in lower survival in unprotected seedlings when compared with protected seedlings (Figure 3B and Figure 4B). In addition, herbivory delayed reproductive development (Table 1). 

Most of the herbivory observed in the two field experiments occurred during the late winter and early spring when ground squirrels (*Urocitellus endemicus*) emerged after a prolonged period of estivation followed by hibernation [41,42]. This timing and the type of cut noted in the seedlings (Figure 3D) strongly suggest that ground squirrels were the primary cause of herbivory. Interestingly, this damage only occurred during the first winter and spring following outplanting. Subsequently, plants showed much less susceptibility to herbivory. These observations suggest changes in plant chemistry or structure that discouraged herbivory. Herbivory may have triggered some of these changes, but they could also have resulted from developmental processes [6,43]. Some observations support this notion; by the second winter in the field, many plants within protectors had branches extending out of them. These branches experienced minimal herbivory. 

While most plants regrew after herbivory, the damage decreased subsequent survival. This decrease occurred in both outplantings, but the pattern and extent of survival decline varied between them. For the first outplanting, most of the mortality happened during the first spring and summer following outplanting. In contrast, for the second outplanting, little mortality occurred during the spring, but mortality became considerable during the summer and continued during the fall and winter. Additionally, twenty-two months after the fall 2018 outplanting, plants without protectors had 51 and 58% lower survival than plants in the plastic and metal protector treatments. Such differences in survival were smaller for the fall 2019 outplanting, where survival for the no-protector treatment was 25 and 37% lower than in the plastic and metal protector treatment. Thus, the capacity to tolerate herbivory was lower in the first than in the second outplanting. 

A possible reason for the differences in herbivory-induced mortality between the two outplantings was the lower precipitation during the spring and early summer of 2019 compared to the same period in 2020 (Figure 1). Due to these differences in precipitation, the onset of drought may have occurred earlier in 2019 than in 2020. Under this scenario, plants that suffered herbivory in 2019 would have had less opportunity to recover than those that experienced herbivory in 2020, leading to earlier and higher mortality. Such an explanation is consistent with the compensatory continuum hypothesis that predicts a decrease in herbivory tolerance under resource-limiting conditions [44]. The reduced tolerance to herbivory in the year with lower precipitation is also in agreement with results in other species, where the ability to regrow and reproduce after herbivory, also known as plant compensation, diminished with less water availability [45,46,47].

In our study, the timing of herbivory and its delayed and negative impact on summer survival allowed us to investigate possible mechanisms by which herbivory reduced drought tolerance. We hypothesized that in browsed plants, the carbon demand for shoot regrowth would occur at the expense of root growth, reducing the plant’s ability to extract water from deeper soil, resulting in lower plant water potentials than uneaten plants. This hypothesis was tested by measuring the predawn and midday water potential during the summer following the 2019 outplanting. Contrary to our hypothesis, we did not detect differences in water potential. During the summer, plants reached water potentials between −2.0 and −8.0 MPa. Values in the upper end of this range, between −2 and −4 MPa, while low, are above those that cause hydraulic failure in *A. tridentata* [48]. In contrast, water potentials below −4 MPa and down to −8 MPa were within a range where significant losses in xylem hydraulic conductivity were likely to occur [48]. The proportion of measured plants reaching a water potential of below −4 MPa was 18% in plants with metal protectors and 25% without protectors, but the difference was not significant (χ^2^ = 0.31). Because values below −4.0 MPa represented only 29% of the Ψ_l_ measured during the summer, a larger sample size may have revealed statistical differences. However, even if this was the case, dissimilarities in Ψ_l_ alone seem insufficient to account for the observed differences in survival.

An additional or alternative factor that may have contributed to the higher mortality of browsed plants is a reduction in their ability to withstand the low Ψ_l_ experienced. Water potentials below −2.5 MPa corresponded with low to minimal stomatal conductance (Figure 5C,D). This decrease in stomatal conductance may have led to periods when plants had a negative carbon balance and depended on NSCs to maintain metabolism [49]. Moderate to severe defoliation often reduce NSC concentrations [13,50,51]. Consequently, a possibility is that plants that suffered herbivory had, before the drought, fewer NSCs than those unbrowsed. Low levels of NSCs can reduce drought tolerance through several effects, such as higher vulnerability to cavitation, impaired ability to osmoregulate and maintain phloem function, and less capacity to recover from xylem embolism after a drought [52,53,54,55]. These effects could have led to higher mortality in unprotected *A. tridentata* seedlings [56]. Non-structural carbohydrates also play an important role in cold tolerance [57,58]. Consequently, fewer NSCs in plants that suffered herbivory could account for their higher mortality during the winter.

Besides its effect on survival, herbivory markedly decreased the percentage of live plants that developed inflorescences in the year of the direct damage and the subsequent year (Table 1). The capacity of plants to compensate for herbivory, resulting in similar or more flower and seed production in browsed than uneaten plants, is highly variable and affected by resource availability [44,59,60]. In our experiments, the timing of herbivory and drought likely prevented a compensatory response. Herbivory occurred in young plants, which, despite resprouting, remained smaller than the protected plants. The loss of vegetative structures combined with drought within a few months of herbivory may have delayed the transition from juveniles to adults and decreased the production of internal signals and photosynthates that promote flowering [61,62,63].

The main differences in survival and flowering occurred between plants without and those with metal or plastic protectors. However, we also detected differences between the metal and plastic protector treatments. In both outplantings, the percentage of plants that experienced herbivory was higher in plants within plastic protectors than in those within metal ones. For the 2018 outplanting, the difference in herbivory between these treatments did not impact survival, but it correlated with a lower percentage of plants developing inflorescences in the plastic treatment. In contrast, for the 2019 outplanting, the higher herbivory in the plastic treatment compared to the metal one was associated with 12% higher survival in the latter but no differences in the percentage of plants developing inflorescences. Thus, in both outplantings, the metal protectors provided some benefits over the plastics ones. Whether these benefits justify using metal over plastic protectors is unclear, but some considerations suggest that the former may have other advantages. In plants smaller than those used in this study, recovery from herbivory may be more difficult. Under these circumstances, metal protectors may have a higher impact on survival. In addition, metal protectors can provide some defense against grasshoppers in years with high herbivory by these insects and are much more durable than plastic ones. Consequently, metal protectors can be used in multiple succeeding outplantings. 

As was noted in the introduction, high seedling mortality is a significant factor hindering the reestablishment of *A. tridentata* in disturbed areas. Based on the mortality caused directly or indirectly by herbivory in this study, practices aimed at reducing it are likely to increase recruitment in *A. tridentata* and thereby contribute to restoring sagebrush habitats. Particularly in habitats where the abundance of ground squirrels or other herbivores is high, the application of protectors seems worth the additional cost associated with their use. However, for large outplantings, the logistics of placing and ultimately removing protectors make their use somewhat impractical [64]. Consequently, developing more efficient methods to reduce herbivory would be valuable. In this regard, one of this study’s results suggests an intriguing possibility. The incidence of herbivory was much lower during the second spring in the field, implying significant developmental or environmental plasticity in plant defenses. Identifying the triggers of this change may provide an opportunity to prime the seedlings before outplanting to reduce their susceptibility to herbivory. Such priming could involve modifying fertilization and watering regimens during late nursery growth to resemble particular field conditions or applying compounds that trigger palatability changes and increase chemical defenses [65,66,67]. 

Independent of the treatment applied, some of the results collected during the summer are informative of seedlings’ physiological characteristics and developmental differences in their ability to cope with drought. The relationship between predawn and midday Ψ_l_ was close to one (Figure 6). Based on the work of Martínez-Vilalta [68], a slope of 1 indicates strict anisohydric stomatal behavior. Anisohydric behavior means that plants keep a relatively constant soil to leaf Ψ gradient as drought develops, allowing them to maintain high *g_s_* and photosynthesis [49,69]. Assuming that the predawn Ψ_l_ represents the water potential of the soil from where the roots took water [70], *A. tridentata* seedlings showed a behavior close to anisohydric. Such anisohydric behavior is consistent with results recently reported for adult plants of the same subspecies; Sharma et al. [71] showed that *A. tridentata* ssp. *wyomingensis* was more anisohydric than *A. tridentata* ssp. *vaseyana*. However, the relationship between the predawn–midday Ψ_l_ gradient and *g_s_* is unclear for our young plants since we did not conduct parallel measurements of these variables. However, based on the midday Ψ_l_ and *g_s_* observed in the following year (Figure 5C,D), it seems very unlikely that there was not a decrease in *g_s_* independent of the predawn–midday Ψ_l_ gradient. In *Eucalyptus gomphocephala* DC., Franks et al. [72] observed a constant plant water potential gradient with increasing water deficits but decreased stomatal conductance. They described this behavior as anisohydric but isohydrodynamic and indicative of parallel *g_s_* and hydraulic conductivity reductions with drought [72]. Whether *A. tridentata* seedlings followed this behavior requires further experimentation [73], but it would explain the apparent discrepancies between water potential gradients and changes in *g_s_* with declining Ψ_l_.

The midday Ψ_l_ also showed less variation and higher values during the summer of 2021 than in the summer of 2020 (Figure 5). For comparable periods (mid-July to late-August), the average midday Ψ_l_ was 1.4 MPa (*p* < 0.0001) higher in 2021 than in 2020. This increase occurred even though the weather was more conducive to drought in 2021 than in 2020. Such results are not entirely unexpected. The plants were larger in 2021 than in 2020 and likely had a more extensive root system to extract water from moister and deeper soil. Nevertheless, the Ψ_l_ data revealed a notable increase in the plant’s ability to maintain higher water potentials with drought, which explains the marked decrease in mortality between the first and second summer in the field.

## 4. Materials and Methods

### 4.1. Plant Material

The plant material used in the experiments was *Artemisia tridentata* ssp. *wyomingensis* seedlings provided by the Bureau of Land Management; this agency uses similar seedlings in restoration projects. Seeds to grow these seedlings had been collected within five miles from our experimental field site at the Morley Nelson Snake River Birds of Prey National Conservation Area (Murphy, ID, USA). The seeds were sown in 150 mL cone-tainers filled with a 3:1 peat moss to vermiculite mix and subsequently grown for ten months before outplanting, as described by Fleege [74].

### 4.2. Experimental Approach

The study involved two similar experiments that started in two consecutive years (2018 and 2019). Both experiments were conducted in adjacent plots in Kuna Butte, ID, USA (43°26′47.32″ N, 116°26′48.61″ W). The soil at this site is Power–McCain silty loam, which is classified as fine–silty, mixed, superactive, mesic Xeric Calciargids [75]. The first experiment started in October 2018. At this time, most of the vegetation at the site was dry and consisted of stalks of non-native plants, mainly crested wheatgrass (*Agropyron cristatum* (L.) Gaertn.), cheatgrass (*Bromus tectorum* L.), and tumble mustard (*Sisymbrium altissimum* L.). We outplanted 750 seedlings in a lattice at a distance of about 1.5 m from each other. The seedlings were randomly assigned to one of three treatments (*n* = 250): without tree protector, with plastic tree protector (25.2 mm mesh, 44 cm height, and 10 cm in diameter), and with metal tree protector (6 mm mesh and closed at the top) (Figure 2). The seedlings were watered immediately after outplanting through a PVC tube inserted about 20 cm from the soil surface, and this watering was repeated two weeks later. After these watering events, the plants only received natural precipitation. A weather station at the site recorded temperature, precipitation, and moisture in the top 20 cm of soil.

The efficacy of the tree protectors in reducing herbivory was assessed by counting the plants that showed significant signs of herbivory, as judged by extensive removal of branches or leaves. These observations, and those of seedling mortality, were made approximately monthly between October 2018 and December 2019 and less frequently during the spring and summer of 2020. In addition, at the end of the 2020 summer, we counted the number of plants bearing inflorescences. 

The second experiment started in October 2019. Conditions at the site were similar to those described earlier. Additionally, we followed identical outplanting methods and protector treatments but with only 150 seedlings per treatment. Seedlings showing herbivory damage and seedling mortality were measured nearly monthly between November 2019 and September 2020 and less frequently during the fall of 2020 and spring and summer of 2021. Plants bearing inflorescences were counted in the late summer of 2020 and 2021. In addition, at the end of summer 2020, we estimated the shoot area for each treatment. For this purpose, we took pictures of 25 randomly selected seedlings per treatment. These photos were used to measure the shoot areas using ImageJ software [76]. 

To assess the effect of herbivory on plant water status, we also measured leaf water potential (Ψ_l_) and stomatal conductance (*g_s_*) in the second experiment. These measurements were only conducted in the no protector and metal protector treatment to reduce work. The determination of midday Ψ_l_ started in the early summer of 2020 and continued to the late summer of 2021. We measured midday Ψ_l_ bi-weekly during the summer and less frequently in fall and spring. In addition, during the summer of 2020, we took measurements of predawn Ψ_l_. Midday and predawn Ψ_l_ measurements were made in eight (2020) or five (2021) plants per sampling day and treatment using a pressure chamber (PMS Instrument Company; Albany, OR, USA). For this purpose, small lateral shoots were wrapped in Saran wrap, excised, and immediately used to determine their Ψ_l_. Stomatal conductance was measured during the summer of 2021 in the same plants used to measure midday Ψ_l_. Three measurements were taken per plant between noon and 2 pm using an SC1 leaf porometer (Meter Group, Pullman, WA, USA).

### 4.3. Data Analyses

The effect of the tree protectors on the number of plants that experienced herbivory and survival was analyzed using the *ggsurvplot* and *pairwise_survdiff* functions in the Survminer R package [77]. To examine the impact of the treatments on the shoot area and the number of plants bearing inflorescence, we used a one-way ANOVA and a chi-square test, respectively. Possible differences in Ψ_l_ and stomatal conductance between the no- and metal protector treatments were evaluated by boxplot comparisons. All statistical analyses were conducted using base functions in R 4.0, except for the boxplots, which were generated with the seaborn library in Python [78,79].

## 5. Conclusions

This study showed that in *A. tridentata,* herbivory by small mammals can markedly increase plants’ susceptibility to abiotic stresses, indirectly limiting the re-establishment and recruitment of this species. Herbivory damage mainly occurred during the first winter and spring following outplanting. Most plants recovered from this damage, but herbivory was associated with higher mortality during summer drought. Based on the water potentials measured, browsed plants were less tolerant of the low water experienced in the summer, presumably resulting in mortality at higher water potentials than unbrowsed seedlings. In addition to its effect on survival, herbivory decreased the percentage of live plants that underwent reproductive development. The causes that determined this reduction require further investigation but are likely linked to the browsed plants’ smaller size and leaf area [61]. Interestingly, herbivory markedly diminished after the first spring in the field. This result strongly suggests that developmental or environmental factors triggered increases in plant defenses or other changes that deter herbivory. Identifying the causes of these changes may allow for the development of more effective approaches to decrease the incidence of herbivory. To maximize the re-establishment of *A. tridentata*, reducing herbivory and its adverse effects on drought tolerance is likely to become more critical due to the expected rise in the frequency and severity of drought associated with climate change [80].

## Figures and Tables

**Figure 1 plants-11-02628-f001:**
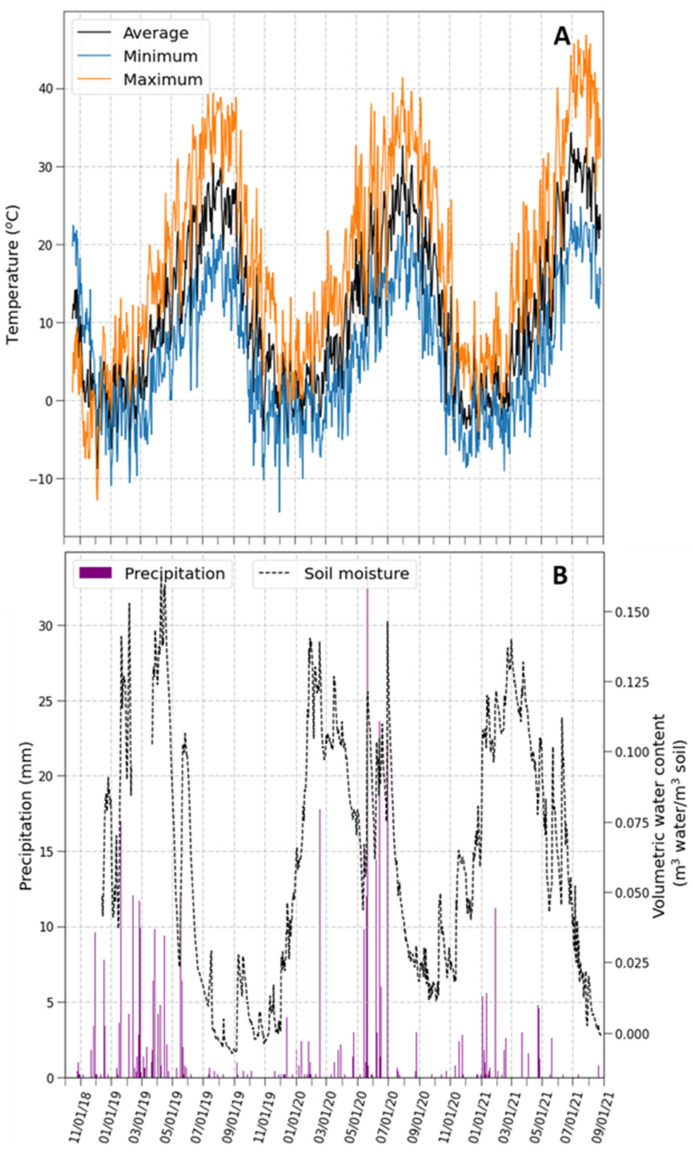
Weather conditions during the experimental period. (**A**) Temperature. (**B**) Soil moisture in the upper 20 cm of the soil and precipitation.

**Figure 2 plants-11-02628-f002:**
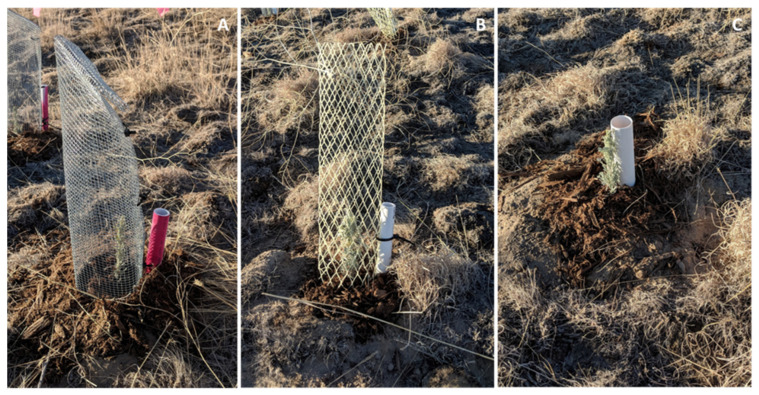
Seedlings after outplanting covered with a metal (**A**) or plastic (**B**) tree protector or without a protector (**C**).

**Figure 3 plants-11-02628-f003:**
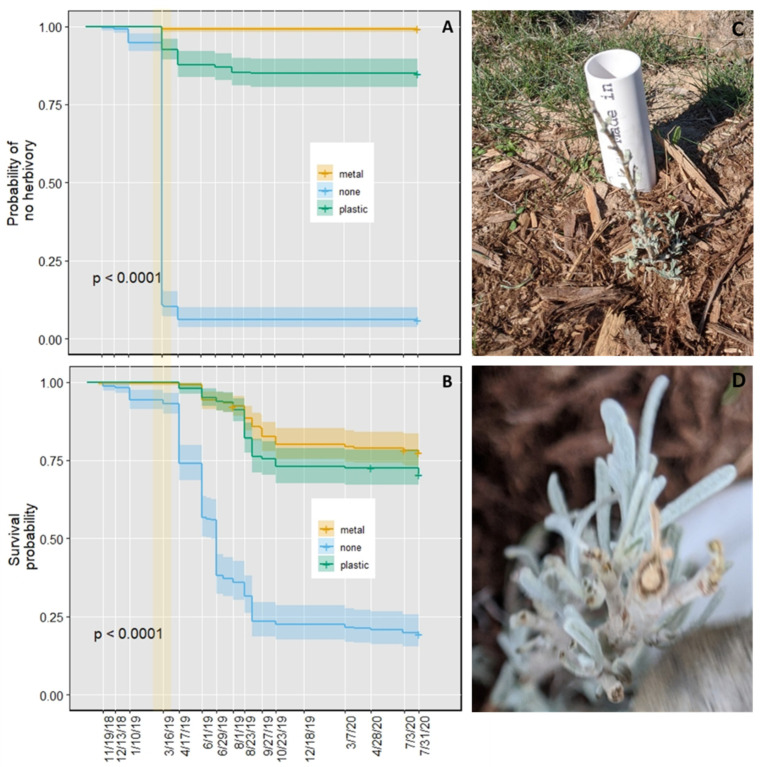
Herbivory damage and survival of *Artemisia tridentata* seedlings outplanted in October 2018. Probability of no herbivory damage (**A**) and survival (**B**) under the different tree protector treatments. Representative picture of herbivory damage in early spring 2019 in plants without protectors (**C**) and a close view of a cut stem (**D**). The shaded area in (**A**,**B**) indicates the period of major herbivory. Curves in (**A**,**B**) are medians and 95% confidence intervals.

**Figure 4 plants-11-02628-f004:**
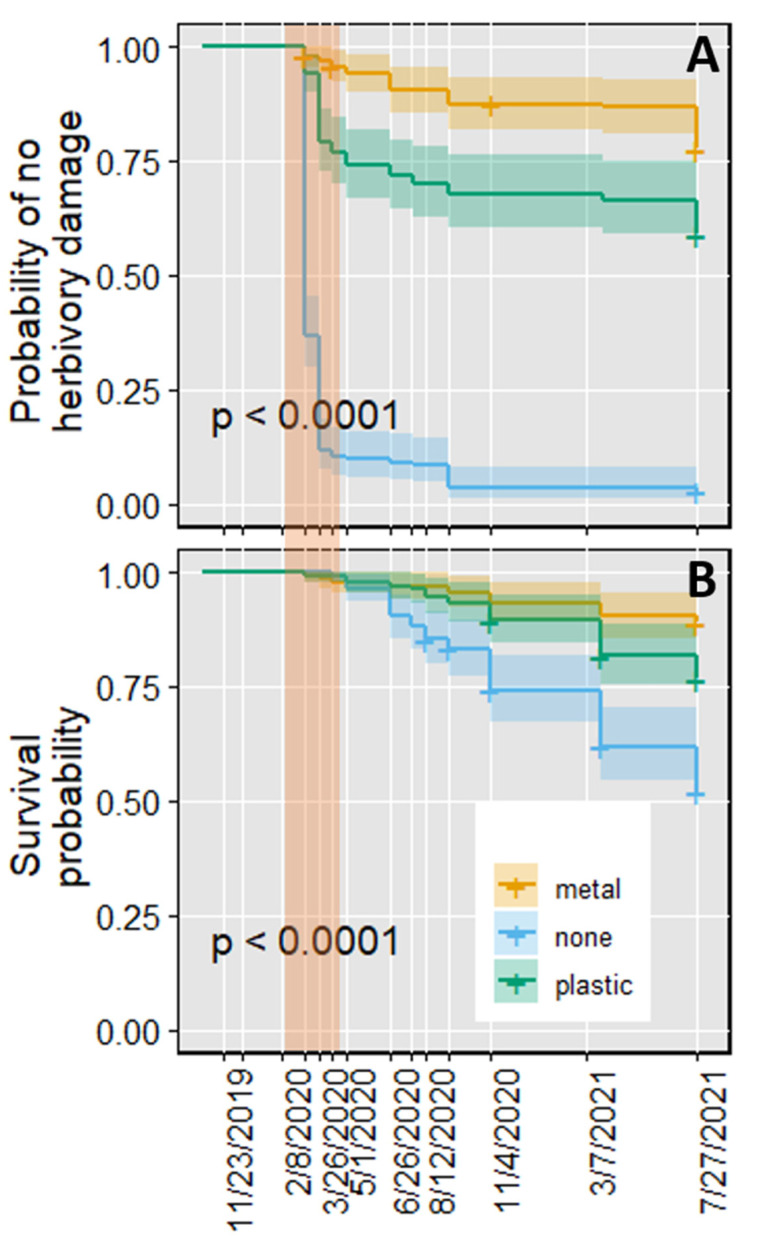
Herbivory damage and survival of *Artemisia tridentata* seedlings outplanted in October 2019. Probability of no herbivory damage (**A**) and survival (**B**) under the different tree protector treatments. The shaded area in (**A**,**B**) indicates the period of major herbivory. Curves in (**A**,**B**) are medians and 95% confidence intervals.

**Figure 5 plants-11-02628-f005:**
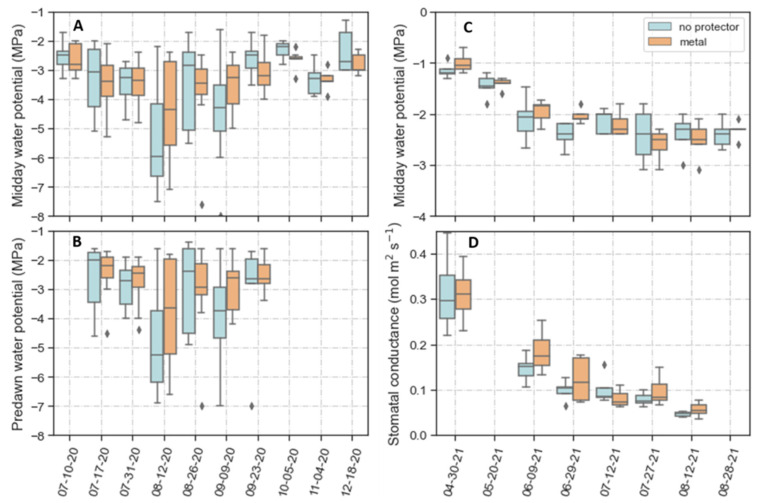
Plant water status of *Artemisia tridentata* seedlings without protectors and within metal protectors during the summer and fall of 2020 (**A**,**B**) and the spring and summer of 2021 (**C**,**D**). Seedlings were outplanted in October 2019. (**A**) Midday water potential. (**B**) Predawn water potential. (**C**) Midday water potential. (**D**) Stomatal conductance. Diamonds indicate outliers. Note: The days when water potentials were measured are plotted as categorical variables rather than a continuous time sequence to make the boxplots more noticeable.

**Figure 6 plants-11-02628-f006:**
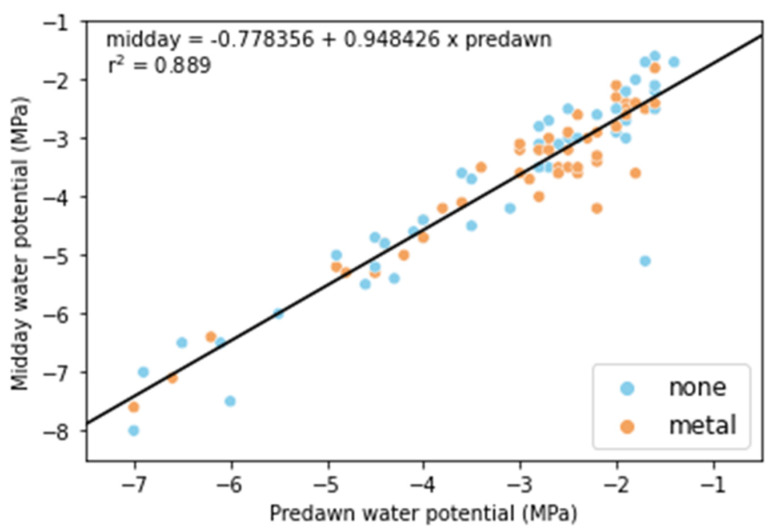
Relationship between predawn and midday water potential in *Artemisia tridentata* seedlings. Measurements were made during the first summer following the 2019 outplanting.

**Table 1 plants-11-02628-t001:** Percentage of *Artemisia tridentata* plants outplanted in October 2019 bearing inflorescence at the end of the 2020, 2021, and 2022 summers. Percentages were calculated based on the number of live plants present during the evaluation of inflorescence occurrence. Values labeled by different letters are significantly different within a column based on χ^2^ tests.

Treatment	2020	2021	2022
no-protector	11 ^b^	12 ^b^	66.6
plastic protector	31 ^a^	41 ^a^	76.9
metal protector	29 ^a^	30 ^a^	80.5

## Data Availability

The data presented in this study are available within the article and its Appendix A.

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
