# Peer review of "Herbivory Amplifies Adverse Effects of Drought on Seedling Recruitment in a Keystone Species of Western North American Rangelands"

_plants, 2022, doi:10.3390/plants11192628_

Round 1

Reviewer 1 Report

This MS presents the results of several field experiments assessing the interactive effects of drought and herbivory on seedlings of Artemesia tridentata, the dominant shrub across much of the Intermountain Western US.  The principal findings are that herbivory exacerbates drought effects by placing seedlings in a bind between competing requirements to ameliorate the effects of the two stresses.  As a basic research project, the findings are of interest, as seedling performance is a critical process in maintaining ecosystem resilience and resistance to the effects of invasive annuals.  From an applied aspect, assessing the multi-year effectiveness of the two protective covers (metal vs plastic) is of clear importance to restoration efforts, as the cost and ease of deploying these types of treatments are considerably different.  Overall, this is a very sound MS; if I have one complaint, it is a minor one: the applied aspects of this study are in many ways more compelling than the basic aspects (sound though they are).  Including a brief paragraph in the discussion regarding the value of this study to restoration/conservation challenges facing sagebrush-steppe ecosystems would certainly add to its value.

Author Response

The reviewer suggested adding a brief paragraph to the discussion to indicate the implications of the results for restoring sagebrush habitats. We added the following paragraph: 

As noted in the introduction, high seedling mortality is a significant factor hindering the reestablishing of A. tridentata in disturbed areas. Based on the mortality caused directly or indirectly by herbivory in this study, practices aimed at reducing it are likely to increase recruitment in A. tridentata and thereby contribute to restoring sagebrush habitats. Particularly in habitats where the abundance of ground squirrels or other herbivores is high, the application of protectors seems worth the additional cost associated with their use. However, for large outplantings, the logistics of placing and ultimately removing protectors make their use somewhat impractical [64]. Consequently, developing more efficient methods to reduce herbivory would be valuable. In this regard, one of this study's results suggests an intriguing possibility. The incidence of herbivory was much lower during the second spring in the field, implying significant developmental or environmental plasticity in plant defenses. Identifying the triggers of this change may provide an opportunity to prime the seedlings before outplanting to reduce their susceptibility to herbivory. Such priming could involve modifying fertilization and watering regimens during late nursery growth to resemble particular field conditions or applying compounds that trigger palatability changes and increase chemical defenses [65–67]. 

The above paragraph is in lines 343 to 358 of the revised manuscript and includes four new references.

Reviewer 2 Report

Climate change is a crucial issue among the serious emerging problems which got global attention in the last few decades. With climate change, worldwide crop production has been seriously affected by drought stress. This study investigated the impact of herbivory on seedlings' drought tolerance for a keystone species in the steppe of western North American rangelands, the shrub Artemisia tridentata. Notably, the authors examined the effects of herbivory in two field experiments where seedlings were without tree protectors or within a plastic or metal-mesh tree protectors. This manuscript is in general well written, logically structured, well-illustrated and easy to understand. It also addresses a subject that is of great interest to the scientific community. The title clearly describes the contents of the paper. The abstract is well written. It encapsulates the entire study (a bit of introduction, aim, result and outcome). The introduction chapter is well written and gives a good background of the research in question. Also, the aim of the study is evident in the beginning and concluding parts. I believe that the Materials and Methods section is well-structured and scientifically sound. The results are well presented, figures and tables are correct. Literature reviews in the discussion section of the manuscript are very professional. My comment is mainly on a relatively minor matter, I think there should be a conclusion chapter at the end of the article, summarizing the key achievements of the work. This comment does not influence a positive impression of the article. 

Author Response

The reviewer suggested adding a conclusions section. In the original manuscript, the discussion's last paragraph was similar to a short conclusion section since we summarized the study's main results. However, we agree with the reviewer that having a separate conclusions section is valuable. Consequently, we deleted the last paragraph of the discussion and moved it to the conclusions after some changes and the incorporation of additional content. The added conclusions are included below.

  1. Conclusions

This study showed that in A. tridentata, herbivory by small mammals can markedly increase the plants' susceptibility to abiotic stresses, indirectly limiting the re-establishment and recruitment of this species. Herbivory damage mainly occurred during the first winter and spring following outplanting. Most plants recovered from this damage, but herbivory was associated with higher mortality during summer drought. Based on the water potentials measured, browsed plants were less tolerant of the low water experienced in the summer, presumably resulting in mortality at higher water potentials than unbrowsed seedlings. In addition to its effect on survival, herbivory decreased the percentage of live plants that underwent reproductive development. The causes that determined this reduction require further investigation but are likely linked to the browsed plants' smaller size and leaf area [61]. Interestingly, herbivory markedly diminished after the first spring in the field. This result strongly suggests that developmental or environmental factors triggered increases in plant defenses or other changes that deter herbivory. Identifying the causes of these changes may allow the development of more effective approaches to decrease the incidence of herbivory. To maximize the re-establishment of A. tridentata, reducing herbivory and its adverse effects on drought tolerance is likely to become more critical due to the expected rises in the frequency and severity of drought associated with climate change [80].